# A Mulberry Diels-Alder-Type Adduct, Kuwanon M, Triggers Apoptosis and Paraptosis of Lung Cancer Cells through Inducing Endoplasmic Reticulum Stress

**DOI:** 10.3390/ijms24021015

**Published:** 2023-01-05

**Authors:** Mengjiao Ma, Xiaoyi Luan, Hao Zheng, Xiaoning Wang, Shuqi Wang, Tao Shen, Dongmei Ren

**Affiliations:** Key Laboratory of Chemical Biology (Ministry of Education), School of Pharmaceutical Sciences, Shandong University, 44 West Wenhua Road, Jinan 250012, China

**Keywords:** mulberry Diels-Alder-type adducts, Kuwanon M, apoptosis, paraptosis, ER stress

## Abstract

The mulberry tree (*Morus alba*) has been cultivated in China for thousands of years. Mulberry Diels-Alder-type adducts (MDAAs) are characteristic constituents of the genus Morus. The unique structure and diverse bioactivities of MDAAs have attracted the attention of researchers. Kuwanon M (KWM) is an MDAA isolated from the root bark of *Morus alba*. This research reports the growth inhibitory effects of KWM on human lung cancer cells and its possible mechanism. In A549 and NCI-H292 cells, KWM treatment induced suppression of cell proliferation and migration. The appearance of chromatin condensation, phosphatidyl serine exposure and caspase cleavage indicated the arising of apoptosis. The loss of mitochondrial membrane potential (MMP), release of cytochrome *c* and dysregulation of Bax/Bcl-2 demonstrated that the KWM-induced apoptosis was through the mitochondrial pathway. Paraptosis was simultaneously detected under KWM treatment, as evidenced by the exhibition of cytoplasmic vacuolation, down-regulation of Alix and up-regulation of endoplasmic reticulum (ER) stress-related proteins. Mechanistically, ER stress induced activation of unfolded protein response (UPR) pathways and activation of the MAPK (JNK and ERK) pathway, all of which were critical for KWM-induced apoptosis and paraptosis. These findings suggested the possibility that KWM might be considered as a potential lung cancer therapeutic agent.

## 1. Introduction

The mulberry tree (*Morus alba*) has been cultivated in China for thousands of years. Mulberry planting and silkworm feeding have been a very important part of Chinese agriculture. Other than agricultural use, mulberry leaves have also been processed into tea and consumed as a healthy drink. The leaves, twig, fruit and root bark of mulberry are all listed in Chinese Pharmacopoeia. The root bark of mulberry is known as “Sang-Bai-Pi” in Chinese medicine, and is considered to possess the function of treating cough, asthma, edema and oliguria [1]. Flavonoids, stilbenes, benzofurans and Mulberry Diels-Alder-type adducts (MDAAs) are the main constituents of the root bark of mulberry [2]. Among these, MDAAs are unique compounds with complicated structures which has made these compounds an attractive research topic in recent decades.

With the progress of diagnosis and treatment, the overall cancer incidence rates and death rates have declined over the past few decades [3]. However, lung cancer is still one of the most common and deadliest types of cancer. It was estimated that almost 25% of all cancer deaths were attributable to lung cancer in 2021 [4,5]. For most types of cancer, survival has improved due to improvement in treatment protocols, including targeted therapy [6]. In contrast, advances for lung cancer survival have been slow because of a certain portion of cases, which are diagnosed at a late stage [7]. Therefore, there is still a demand to develop novel chemotherapeutic drugs to improve the survival of lung cancer patients.

Mulberry Diels-Alder adducts (MDAAs) are characteristic compounds isolated from Moraceous plants. Generally, this type of compound is biosynthetically derived from [4+2]-cycloaddition of chalcones (dienophiles) and prenylated phenols (dienes) [8]. MADDs have been found to possess a variety of bioactivities, including anti-oxidation [9], PTP1B inhibition [10,11], cytotoxicity [12], neuro-protection [13] and anti-inflammation [14]. Kuwanon M (KWM) is an MDAA purified from the root bark of *Morus alba* by our group. The biosynthesis of KWM was proposed through Diels-Alder cycloaddition with the C-8 prenyls of two Kuwanon C as both dienophile and diene (Figure 1A). During a screening to identify anticancer agents from natural compounds, KWM was found to have good cytotoxic activity against human lung cancer cells. Therefore, studies on the anti-proliferation effects of KWM against lung cancer cells and its possible mechanism were conducted.

Apoptosis is the most common and well elucidated form of programmed cell death (PCD). Induction of apoptosis is considered as the major mechanism of most chemotherapeutic agents [15]. Therefore, typical cancer therapies mainly focus on promoting cancer cell elimination by induction of apoptosis [16]. In recent years, some non-apoptotic forms of PCD such as autophagy, paraptosis, ferroptosis, methuosis and necroptosis have been identified. Targeting these non-canonical cell death modes provides new options for cancer treatment [17]. Paraptosis is a cell death mode characterized by displaying mitochondria and/or endoplasmic reticulum (ER) dilation [18]. Recent studies have demonstrated that some natural compounds suppressed cancer cells proliferation through triggering paraptotic cell death [19,20].

In this study, we report, for the first time, that KWM-induced human non-small cell lung cancer (NSCLC) cell death through both apoptosis and paraptosis. ER stress induced activation of unfolded protein response (UPR) pathways and activation of the MAPK (JNK and ERK) pathway, all of which were mechanistically involved in KWM-induced apoptosis and paraptosis.

## 2. Results

### 2.1. The Structure and Purity Determination of KWM

The structure of KWM (Figure 1A) was determined by MS, ^1^H NMR, ^13^C NMR and 2D NMR measurement. Spectroscopic data was also compared with data previously reported in the literature [21]. The MS spectrum of KWM showed the quasi-molecular ion peak at *m*/*z* 841.7, which suggested the molecular formula was C_50_H_48_O_12_. The purity was 96%, as detected by HPLC.

### 2.2. KWM Inhibited the Growth and Migration of Lung Cancer Cells

An MTT assay was used to evaluate the proliferation inhibition effects of KWM in two kinds of NSCLC cell lines, A549 and NCI-H292. The results indicated that KWM suppressed the cell viability of A549 and NCI-H292 cells in dose- and time-dependent manners (Figure 1B). The IC_50_ values of KWM treatment for 72 h were 11.79 μM on A549 and 8.98 μM on NCI-H292 cells, respectively. An immortalized normal human lung epithelial cell line, BEAS-2B, was selected to evaluate the selective cytotoxicity of KWM. As shown in Figure 1B, cell viability decreased after 48 h treatment with KWM in BEAS-2B cells, but the degree of reduction was much slighter than that in NSCLC cells. In addition, a colony formation assay was carried out for further evaluation of the anti-proliferative effects of KWM. After 12 days of treatment with 5 μM and 10 μM of KWM, the colony formation ability of A549 and NCI-H292 cells was significantly suppressed (Figure 1C). Furthermore, a scratch wound-healing assay was performed to assess the anti-migration effects of KWM. As shown in Figure 1D, KWM remarkably inhibited the healing of the scratches, indicating that this compound possessed the ability to reduce lung cancer cell migration.

### 2.3. KWM Caused Caspase-Dependent Apoptotic Cell Death in Lung Cancer Cells

In order to elucidate whether KWM-induced cell death was associated with apoptosis, DAPI labeling and Annexin V-FITC/PI double staining were carried out. As shown in Figure 2A, more condensed and fragmented DAPI-stained nuclei were observed in KWM-treated cells than in control cells and a higher proportion of Annexin V-FITC positive cells appeared in KWM-administrated groups than in the untreated group in flow cytometry assay (Figure 2B). These results suggested the occurrence of apoptosis after KWM treatment in A549 and NCI-H292 cells. As a molecular marker of apoptosis, the cleavage of PARP was detected by immunoblot analysis, represented by the increased protein level of cleaved PARP and decreased protein level of PARP (Figure 2C), further demonstrating the induction of apoptosis by KWM treatment. Moreover, KWM-induced apoptosis was revealed to be involved in the activation of caspase cascades. As shown in Figure 2C, the expression of cleaved caspase-9 and -3 increased while pro caspase-9 and -3 decreased after KWM treatment. A pan-caspase inhibitor Z-VAD-FMK partially reversed the cell viability reduction induced by KWM (Figure 2D), which confirmed that KWM caused caspase-dependent apoptosis in the two kinds of lung cancer cell lines.

### 2.4. KWM Triggered Cell Cycle Arrest and Accumulation of ROS in Lung Cancer Cells

In order to evaluate the actions of KWM on cell cycle distribution, A549 and NCI-H292 cells were administrated with different concentrations of KWM for 24 h and then measured by FACS. The results indicated that KWM treatment increased the population at the G_0_/G_1_ phase, while decreasing the percentage of cells at the S phase, accordingly (Figure 3A). This result suggested that KWM blocked the cell cycle at the G_0_/G_1_ phase in both A549 and NCI-H292 cells.

Generally, excessive production of intracellular ROS is related to apoptosis [22,23]. The ROS levels induced by KWM treatment were determined by loading with DCFH-DA and analyzed by flow cytometry. The results indicated that the KWM treatment significantly increased the production of intracellular ROS in both A549 and NCI-H292 cells (Figure 3B).

### 2.5. KWM-Induced Apoptosis through Mitochondria-Mediated Intrinsic Pathway

Mitochondria play a key role in mammalian cell apoptosis. The respiratory chain of mitochondrial inner membrane is the main source of ROS; meanwhile, mitochondria are the proapoptotic target of excess ROS. Overproduction of ROS may cause mitochondrial membrane depolarization, release of cytochrome *c*, activation of caspase cascades, and, eventually, induce apoptosis [24]. To investigate the effects of KWM on mitochondrial function, A549 and NCI-H292 cells were treated with KWM, then MMP, cytochrome *c* release and the expression of apoptosis-related proteins were analyzed. MMP was detected by JC-1 staining first. As shown in Figure 4A, the control group showed a high red to green fluorescence intensity ratio in both cell lines, while KWM treatment decreased the ratios dramatically. Carbonyl cyanide 3-chlorophenylhydrazone (CCCP) treated cells were used as positive control, which showed a very small red to green ratio. Thus, the JC-1 staining results proposed that KWM initiated depolarization of the mitochondrial membrane and decreased MMP in A549 and NCI-H292 cells. Another detection method, flow cytometric analysis, revealed consistent results (Figure 4B).

Once mitochondrial membrane depolarization occurred and MMP collapsed, cytochrome *c* would be released from the intermembrane of the mitochondria into the cytoplasm [25]. KWM treatment induced translocation of cytochrome *c*, which was proved by examination of the cytochrome *c* protein level in the cytoplasm and the mitochondria, respectively. As shown in Figure 4C, a distinct upregulation of cytochrome *c* in the cytosol fraction, and corresponding downregulation in the mitochondria, were observed in both A549 and NCI-H292 cells exposed to KWM for 24 h.

Cytochrome *c* release usually implies the early event of apoptosis. The Bcl-2 protein family regulates the translocation of cytochrome *c* and loss of MMP [26]. The expression of Bcl-2 family of proteins, including Bax, Bcl-2 and Bcl-xL was analyzed by western blot in A549 and NCI-H292 cells. As shown in Figure 4D, KWM treatment increased the protein levels of the pro-apoptotic protein, Bax, and decreased the levels of anti-apoptotic proteins, Bcl-2 and Bcl-xL. Together with the above results, this suggested that KWM induced apoptosis through the intrinsic mitochondrial pathway in A549 and NCI-H292 cells.

### 2.6. KWM Triggered Paraptosis in Lung Cancer Cells

During the cytotoxicity assay, a phenomenon was observed whereby KWM induced the accumulation of cytoplasmic vacuoles in both A549 and NCI-H292 cells (Figure 5A). As cytoplasmic vacuoles have been reported as characteristics of paraptosis, we assumed that KWM induced paraptosis in lung cancer cells. Paraptosis is a non-apoptotic programmed cell death mode that is insensitive to apoptosis inhibitors. As mentioned above, pan-caspase inhibitor Z-VAD-FMK pretreatment reversed the cell viability reduction obviously, but not completely (Figure 2D), and the cytoplasmic vacuolation was not affected (Figure 5B), suggesting the co-existence of apoptosis and paraptosis in KWM-treated cells.

To investigate the origin of cytoplasmic vacuoles induced by KWM, cells were stained with ER-Tracker red and Mito-Tracker red, which could label endoplasmic reticulum (ER) and mitochondria, respectively. It was observed that both ER fluorescence and mitochondrial fluorescence were colocalized with vacuoles (Figure 5C), suggesting that KWM-induced vacuoles originated from ER dilatation and mitochondrial swelling, which are exactly the characteristics of paraptosis.

Next, other features of paraptosis were examined in KWM-treated A549 and NCI-H292 cells. The decreased expression of ALG-2-interacting protein X (Alix) was recognized as a molecular marker of paraptosis [27]. As shown in Figure 5D, the results of the immunoblot assay indicated a dose-dependent suppression of Alix expression in KWM treated A549 and NCI-H292 cells. 

Mitogen-activated protein kinases (MAPKs) activation is supposed to accompany the occurrence of paraptosis [27]. The activation of JNK and ERK 1/2, two subfamilies of MAPKs, was examined after treatment with KWM in NCI-H292 and A549 cells. As indicated in Figure 5D, KWM treatment increased the levels of both p-JNK and p-ERK 1/2. The functional significance of JNK and ERK 1/2 activation in KWM-induced paraptosis was also investigated. As shown in Figure 5E,F, pretreatment cells with JNK and ERK inhibitors, SP600125 and U0126, blocked KWM-induced cytoplasmic vacuolization almost completely and blocked cell death to some extent. Based on these results, it is suggested that paraptosis may contribute to KWM-induced cell death in lung cancer A549 and NCI-H292 cells.

### 2.7. KWM Induced ER Stress in Lung Cancer Cells

The generation of ER dilation usually leads to persistent ER stress [28]. Therefore, proteins associated with ER stress were detected by immunoblot analysis in KWM-treated A549 and NCI-H292 cells. GRP78 is an ER chaperone whose expression is induced during ER stress. So, GPR78 was detected firstly in A549 and NCI-H292 cells, as indicated in Figure 6A, GPR78 was upregulated by exposure to KWM. ER stress can always result in the activation of a signal transduction pathway, unfolded protein response (UPR) signaling. UPR is usually initiated by three ER transmembrane sensors: IRE1, PERK and ATF-6 [29]. The protein levels of IRE1α, p-PERK and ATF-6 were then detected. As shown in Figure 6A, the expression of p-PERK, IRE1α and ATF6 were enhanced dose-dependently under KWM treatment, demonstrating the activation of UPR signaling. Accordingly, as target genes of the three UPR stress sensors, ATF-4, p-eIF2α and GADD153 were activated coincidently, reflected by increased protein levels under KWM administration (Figure 6A). 

The expression of several representative proteins, including eIF2α, PERK and GADD153, was further tested at various time points. KWM treatment upregulated GADD153 expression and eIF2α and PERK phosphorylation in a time-dependent manner (Figure 6B). The higher levels of the detected proteins were sustained for up to 24 h, indicating the occurrence of prolonged ER stress.

### 2.8. ER Stress Contributed to KWM-Induced Apoptosis and Paraptosis in A549 and NCI-H292 Cells

Accumulating evidence suggests that ER stress is implicated in both apoptosis and paraptosis [30,31]. Cycloheximide (CHX) is an inhibitor of protein synthesis. Treatment of cells with CHX can decrease the protein burden on ER and, thus, ameliorate effects of ER stress, which can, therefore, reduce ER stress-mediated cell death [32]. The acid, 4-phenylbutyric acid (4-PBA), serves as an ER stress inhibitor through assisting protein folding and preventing misfolded protein aggregation [33]. To evaluate the relevance of KWM-induced ER stress to apoptosis and paraptosis, CHX and 4-PBA were used to pretreat A549 and NCI-H292 cells. As shown in Figure 7A,B, both cytoplasmic vacuolation and cell death induced by KWM were effectively alleviated in the presence of CHX and 4-PBA. Furthermore, KWM-induced upregulation of GRP78 and GADD153, three representative proteins of ER stress, was blocked by pretreatment of CHX and PBA (Figure 7C). KWM-induced cleavage of PARP and caspase-3, was also alleviated by CHX and 4-PBA (Figure 7D). These results collectively implied that ER stress contributed to KWM-induced apoptosis and paraptosis.

## 3. Discussion

Substantive literature indicates that MDAAs are structurally unique natural compounds from Moraceous plants. *Morus alba*, commonly referred to as mulberry, has been widely cultivated in China for thousands of years. The root bark of *M. alba*, named Sang-Bai-Pi in Chinese, is included in Chinese Pharmacopoeia as a traditional Chinese medicine for the treatment of cough, asthma, edema and oliguria. MDAAs, a group of rare polyphenols derived from [4+2]-cycloaddition of chalcones and prenylated phenols, were revealed as prototypical constituents isolated from the root bark of *M. alba*. There are about 80 MDAAs obtained from Moraceous plants, of which about 30 MDAAs were from the root bark of *M. alba*. MDAAs are reported to have a variety of biological activities, including antioxidant, anti-tumor, anti-inflammatory and neuroprotective effects [34,35]. KWM is an MDAA obtained from the root bark of *Morus alba* by our group. This compound is a Diels-Alder cycloaddition product of two molecules of kuwanon C. The complicated structure and the preliminary activity screening results of KWM prompted us to conduct further research.

Apoptosis is the main form of programmed cell death. Most chemotherapeutic drugs currently used in clinics exert their anti-tumor effects by inducing apoptosis. MDAAs have been reported to possess cytotoxicity and cancer cell apoptosis inducing effects [36]. In the present study, KWM was demonstrated to induce apoptosis in human lung cancer A549 and NCI-H292 cells (Figure 2A,B). It is recognized that there are two main apoptotic pathways: the extrinsic and the intrinsic pathways. The apoptosis triggered by KWM was elucidated as occurring through the intrinsic pathway, and substantive evidence was included, as follows: the loss of MMP, release of cytochrome *c* into the cytoplasm, the cleavage of caspase-9 and caspase-3, an increased level of Bax and a decreased level of Bcl-2 (Figure 2C and Figure 4).

Accumulating evidence revealed that the resistance to chemotherapy of most cancer cells is because of acquiring the ability to escape apoptosis. Therefore, simply inducing apoptosis is not sufficient to exert the efficacy of chemo-drugs. Discovering alternative agents that target the non-apoptosis cell death mode might be a productive strategy to conquer chemoresistance. Paraptosis is a non-apoptotic form of programmed cell death and characterized by vacuolation of cytoplasm derived from swollen ER and/or mitochondria. Distinct from apoptosis, paraptosis is a lack of response to caspase inhibitors and Bcl-xL. Growing evidence established that inducing paraptosis is an alternative way to overcome the chemoresistance of cancer cells [37,38]. The current study found that KWM-induced apoptosis was accompanied by cytoplasmic vacuolization, which was proven to arise from the swollen ER and mitochondria, which are the distinguishing features of paraptosis (Figure 5A,C). The down-regulation of Alix and up-regulation of p-JNK and p-ERK were also consistent with paraptosis, indicating KWM initiated paraptosis together with apoptosis (Figure 5D).

The ER is a protein handling, modification and folding factory inside cells. Normally, the protein-folding process is exquisitely regulated, but when external factors and intracellular events disrupt this process, ER stress is provoked, and misfolded or unfolded proteins accumulate in the ER lumen. Subsequently, UPR which is an adaptative mechanism able to restore ER homeostasis, is induced. Persistent, unresolved ER stress can cause cell death [39]. KWM induced ER stress and consequent UPR were evidenced by the following: activation of ER stress marker proteins, including PERK, ATF6 and IRE1α (Figure 6A,B); activation of ER chaperone GRP78 and ER stress related protein GADD153 (Figure 6A,B); and CHX and 4-PBA reversing the production of cytoplasmic vacuoles (Figure 7A,B). Furthermore, ER stress contributed to KWM-induced apoptosis and paraptosis in A549 and NCI-H292 cells, demonstrated by CHX and 4-PBA alleviating the toxicity of KWM. The cleavage of apoptosis marker proteins and the activation of ER stress representative proteins were both blocked (Figure 7C,D).

Collectively, KWM induced two distinct kinds of cell death modes simultaneously in human lung cancer cells, which amplify the anti-proliferation effects of KWM in cancer cells. ER stress and its consequential UPR contributed to both apoptosis and paraptosis.

## 4. Materials and Methods

### 4.1. Cell Culture

A549 (originated from adenocarcinoma) and NCI-H292 (mucoepidermoid cancer), two human lung cancer cell lines, and an immortalized normal human lung epithelial cell line BEAS-2B, were obtained from Shanghai Cell Bank (Shanghai, China). The basic medium for A549 and NCI-H292 culture was RPMI (Roswell Park Memorial Institute) 1640 (Gibco, Grand Island, NY, USA), for BEAS-2B was DMEM (Dulbecco’s Modified Eagle Medium, Gibco, Grand Island, NY, USA). All culture media were supplemented with 10% (*v*/*v*) FBS (fetal bovine serum) (HyClone, Logan, UT, USA), 100 U/mL penicillin and 100 U/mL streptomycin. All cell lines were maintained at 37 °C with 5% CO_2_ in a humidified incubator.

### 4.2. Chemicals, Reagents and Antibodies

Kuwanon M was separated from the root bark of *Morus alba*. Thiazolyl blue tetrazolium bromide (MTT), dimethyl sulfoxide (DMSO) and tunicamycin were purchased from Solarbio Life Sciences (Beijing, China). 4, 6-dianmidino-2-phenylindole (DAPI) and 2′, 7′-dichlorodihydrofluorescein di-acetate (DCFH-DA) were purchased from Genview Scientific Inc. (Tallahassee, FL, USA). Z-VAD(OMe)-FMK, cisplatin (CDDP), cycloheximide (CHX) and 4-phenylbutyric acid (4-PBA) were bought from MedChemExpress (Trenton, NJ, USA). SP600125 and U0126 were purchased from Selleckchem (Houston, TX, USA). 5, 5′, 6, 6′-tetrachloro-1, 1′, 3, 3′-tetraethyl benzimidazolyl-carbocyanine iodide (JC-1) was purchased from Cayman Chemical (Ann Arbor, MICH, USA). Mito-Tracker Red was obtained from Life Technologies (Waltham, MA, USA). ER-Tracker Red and carbonyl cyanide 3-chlorophenylhydrazone (CCCP) were purchased from Beyotime Biotechnology (Shanghai, China).

Primary antibodies against Caspase-9 (Cat# 10380-1-AP), Caspase-3 (Cat# 19677-1-AP), cytochrome *c* (Cat# 10993-1-AP), Bax (Cat# 60267-1-Ig), Bcl-2 (Cat# 12789-1-AP), Bcl-xL (Cat# 10783-1-AP), β-tubulin (Cat# 10068-1-AP), GAPDH (Cat# 60004-1-Ig) and β-actin (Cat# 66009-1-Ig) were purchased from Proteintech Group (Wuhan, Hubei, China). Primary antibodies against PARP (Cat# 9542), cleaved caspase-3 (Cat# 9661), cleaved caspase-9 (Cat# 9505), and ATF4 (Cat# 11815) were purchased from Cell Signaling Technology (Beverly, MA, USA). Primary antibodies against p-PERK (Cat# AP0886) and p-eIF2α (Cat# AP0692) were purchased from AB clonal (Wuhan, Hubei, China). Primary antibodies against GRP78 (Cat# sc-166490), IRE1α (Cat# sc-390960), eIF2A (Cat# sc-517214), ATF-6α (Cat# sc-166659), GADD153 (Cat# sc-7351), Alix (Cat# sc-53540), JNK (Cat# SC-7345), p-JNK (Cat# sc-6254), ERK (Cat# sc-514302) and p-ERK (Cat# SC-7383) were purchased from Santa Cruz Biotechnology (Dallas, Texas, USA). Anti-rabbit IgG (Cat# SA00001-2) and anti-mouse IgG (Cat# SA00001-1) were from Proteintech Group (Wuhan, Hubei, China).

### 4.3. Isolation and Identification of Kuwanon M

Kuwanon M was separated from the root bark of *Morus alba* mainly by multiple chromatographic methods. In brief, air-dried and powdered root bark of *M. alba* (10.0 kg) were extracted with EtOH-H_2_O (80:20, *v*/*v*, 20 L × 3) under reflux. After concentration, the crude extract (600 g) was suspended in H_2_O and partitioned with Petroleum Ether (PE), EtOAc and n-BuOH successively. The EtOAc fraction was condensed to give a residue (150 g), which was next subjected to silica gel column chromatography (CC) using PE-EtOAc (100:0–0:100, *v*/*v*) as eluent to afford five major fractions (CH1–CH5). Fraction CH3 (12 g) was further separated by silica gel CC, followed by Sephadex LH-20 CC (CH_2_Cl_2_-MeOH, 1:1) to produce KWM (108 mg). The purity of KWM was determined by HPLC using an Agilent Zorbax Eclipse XDB-C_18_ (150 × 4.6 mm I.D., 5 μm) column, acetonitrile-H_2_O (60:40, *v*/*v*) was used as mobile phase, the flow rate was 1.0 mL/min at room temperature and the detection wavelength was set at 265 nm.

^1^H NMR and ^13^C NMR data of KWM were listed below. ^1^H NMR (600 MHz, acetone-*d_6_*) δ ppm: 6.01 (1H, s, H-6), 6.44 (1H, d, J = 3.3 Hz, H-3′), 6.38 (1H, dd, J = 9.0, 3.3 Hz, H-5′), 7.07 (1H, d, J = 9.0 Hz, H-6′), 3.10 (2H, m, H_2_-9), 5.12 (1H, m, H-10), 3.87 (1H, s, H-14), 5.39 (1H, brs, H-15), 1.47 (1H, m, H_a_-18), 2.17 (1H, m, H_b_-18), 2.74 (1H, m, H-22), 5.65 (1H, d, J = 10.0 Hz, H-23), 6.38 (1H, s, H-6′’), 6.57 (1H, d, J = 3.3 Hz, H-3″′), 6.53 (1H, dd, J = 9.0, 3.3 Hz, H-5″′), 7.31 (1H, d, J = 9.0 Hz, H-6″′), 3.19 (2H, m, H_2_-9″), 5.14 (1H, m, H-10″), 1.57 (9H, s, 12/12′/17-CH_3_), 1.42 (3H, s, 13-CH_3_), 1.43 (3H, s, 13′-CH_3_), 0.67 (3H, s, 20-CH_3_), 0.97 (3H, s, 21-CH_3_).

^13^C NMR (150 MHz, acetone-*d_6_*) δ ppm: 160.4 (C-2), 121.5 (C-3), 183.3 (C-4), 105.7 (C-4a), 157.3 (C-5), 98.7 (C-6), 162.7 (C-7), 103.9 (C-8), 163.1 (C-8a), 112.5 (C-1′), 161.8 (C-2′), 103.6 (C-3′), 161.4 (C-4′), 107.9 (C-5′), 132.1 (C-6′), 24.7 (C-9), 122.5 (C-10), 132.9 (C-11), 25.8 (C-12/C-12″), 17.7 (C-13/C-13″), 31.2 (C-14), 122.8 (C-15), 132.3 (C-16), 23.5 (C-17), 40.9 (C-18), 33.1 (C-19), 28.7 (C-20), 30.4 (C-21), 30.6 (C-22), 71.6 (C-23), 157.3 (C-2″), 121.8 (C-3″), 183.1 (C-4″), 104.9 (C-4a″), 155.8 (C-5″), 98.6 (C-6″), 162.9 (C-7″), 108.1 (C-8″), 160.7 (C-8a″), 113.0 (C-1″′), 157.1 (C-2″′), 104.0 (C-3″′), 161.6 (C-4″′), 108.2 (C-5″′), 132.5 (C-6″′), 24.6 (C-9″), 122.7 (C-10″), 132.0 (C-11″).

### 4.4. MTT Assay

This assay was used for the measurement of cell viability. Briefly, cells (8 × 10^3^ cells/well) were seeded into 96-well plates, and, after adhesion, cells were treated with DMSO or KWM for a certain time. Then, 10 μL of MTT solution (5.0 mg/mL) was added into each well, and incubated for an additional 4 h. Subsequently, the medium was removed and DMSO (100 μL/well) was added to dissolve the formazan. The optical density was measured by a microplate reader (Biorad, Model 680, Hercules, CA, USA) [40].

### 4.5. Wound-Healing Assay

Cells were seeded in 6-well plates with a density of 5 × 10^5^ cells per well and grew overnight. The wounds were made by scratching cell surface straightly with a 10 μL pipette tip. After washing off the exfoliated cells with PBS, cells were then cultured using medium containing 2% FBS with or without KWM. Pictures were acquired using a phase-contrast microscope (Olympus, IX71, Tokyo, Japan). The wound healing rate was calculated as: (area of original wound-area of wound at different time points)/area of original wound × 100% [41].

### 4.6. Apoptosis Assay

Apoptosis was detected by flow cytometry using an Annexin V-FITC/PI (propidium iodide) kit (BD Biosciences, San Jose, CA, USA). Following the instruction of the kit, cells (4 × 10^5^ cells/well) treated with KWM were collected, washed and stained with annexin V-FITC and PI. Stained cells were analyzed using a FACS Calibur flow cytometer (Becton Dickinson, Franklin Lakes, NJ, USA). CDDP treated cells were served as positive control [42].

### 4.7. DAPI Staining

Briefly, cells were treated with or without KWM for 24 h, and, after washing and fixation, cells were incubated with 2.5 μg/mL DAPI solution for 10 min, and then the stained nuclei were recorded by a fluorescence microscope system (Olympus, IX71, Tokyo, Japan) [43].

### 4.8. Intracellular Reactive Oxygen Species (ROS) Measurement

After treatments with KWM for 24 h, cells were washed and incubated with DCFH-DA (10 μM), harvested and cell density adjusted with PBS, analyzed with a FACS Calibur flow cytometer (Becton Dickinson, Franklin Lakes, NJ, USA) at emission wavelength of 530 nm and excitation wavelength of 488 nm [44].

### 4.9. Cell Cycle Analysis

After treatment with KWM for 48 h, cells were collected, washed and fixed in 70% ethanol at 4 °C. Next, cells were stained with 50 μg/mL PI for 30 min following incubation with RNases A (Invitrogen, Waltham, MA, USA) solution (200 μg/mL). Cell cycle was analyzed with a FACS Calibur flow cytometer system (Becton Dickinson, Franklin Lakes, NJ, USA) [43].

### 4.10. Mitochondrial Membrane Potential (MMP) Measurement

Cells were treated with KWM for a certain time, changed to medium with 2 μM JC-1 and incubated for 30 min at 37 °C. Two methods were used for the detection of MMP. First, cells were harvested, resuspended and analyzed by a FACS Calibur flow cytometer (Becton Dickinson, Franklin Lakes, NJ, USA). The second method was to directly detect MMP using a fluorescence microscope. CCCP treatment was used as positive control [45].

### 4.11. Colony Formation Assay

Cells (500 cells/well) were seeded in 6-well plates, and were cultured in medium with and without KWM for two weeks. Afterwards, cells were washed twice with PBS, fixed in cold methanol and stained with 0.1% crystal violet (CV) at room temperature for 1 h, washed and photographed [46].

### 4.12. Western Blot Analysis

Cells after KWM treatments were harvested and lysed with a sample buffer containing 50 mM Tris-HCl [pH 6.8], 2% SDS, 10% Glycerol, 100 mM DTT and 0.1% bromophenol blue. Cell lysates, with equal amounts of proteins, were loaded and electrophoresed on SDS-PAGE gel. Afterwards, proteins were transferred onto the nitrocellulose membrane (Millipore, Boston, MA, USA), and then the membranes were blocked with 5% non-fat milk, incubated with primary antibodies (1:1000) overnight at 4 °C. After that, the membranes were incubated with horseradish peroxidase (HRP)-conjugated secondary antibodies for 1 h at room temperature. Signals were visualized using enhanced chemiluminescence (ECL) reagents. GAPDH, β-tubulin and β-actin were used as internal controls [42].

### 4.13. Live Cell Staining

Cells were seeded and cultured in plate wells with cover-glass at the bottom, and, after treatment with KWM, cells were washed and incubated with Mito-Tracker or ER-Tracker, according to the protocol of the manufacturers. Images were collected by confocal microscope (LSM 700, Zeiss, Oberkochen, Germany) [41].

### 4.14. Statistical Analysis

Results are presented as mean ± standard deviation (SD). To determine the significant difference between two groups, one way analysis of variance (ANOVA) test and post hoc multiple comparison Bonferroni test were used and *p* < 0.05 was considered to be statistically significant.

## 5. Conclusions

In conclusion, KWM derived from *Morus alba* was identified as an inducer of apoptosis and paraptosis. ER stress and UPR play key roles in KWM-induced apoptosis and paraptosis. Based on our findings, we suggested a working model of KWM, illustrated in Figure 8. Our data suggested that KWM could be considered as a potential anticancer agent in future cancer therapeutics.

## Figures and Tables

**Figure 1 ijms-24-01015-f001:**
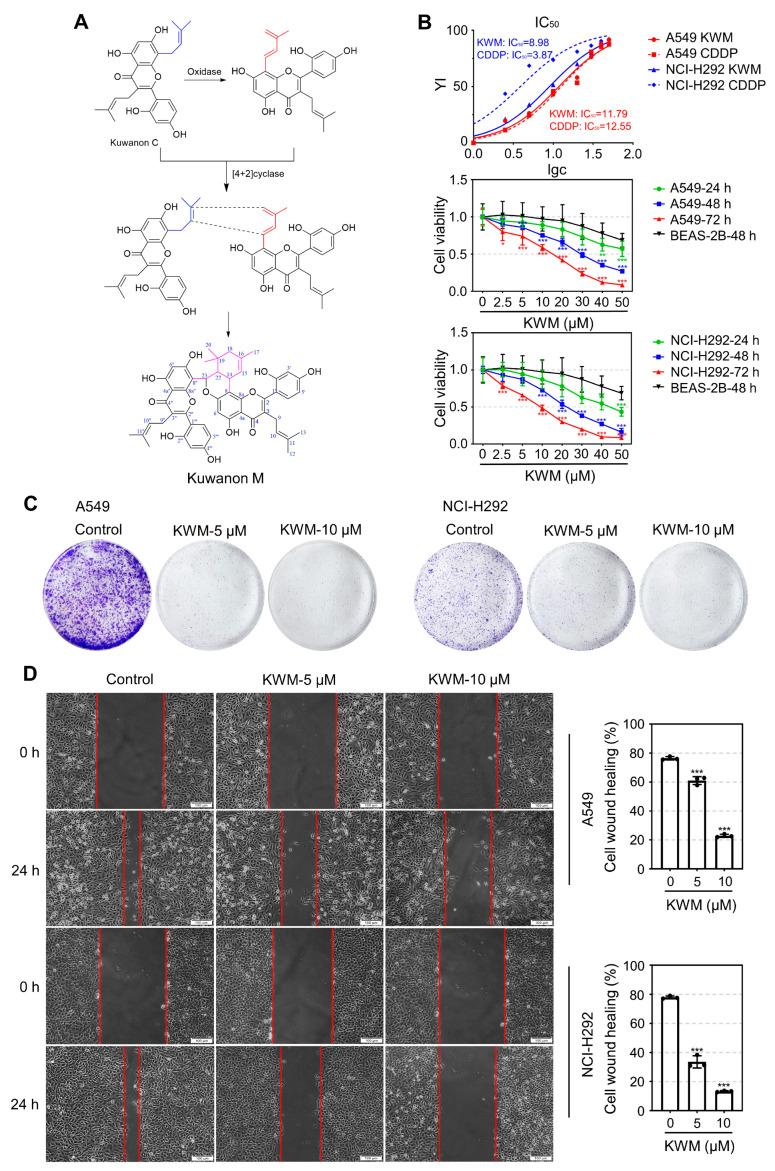
KWM suppressed the proliferation, colony formation and migration of NSCLC cells. (**A**) The chemical structure of KWM. (**B**) BEAS-2B, A549 and NCI-H292 cells were exposed to KWM with indicated concentrations for 24, 48 and 72 h, cell viability was detected by MTT assay, cisplatin was used as positive control. (**C**) Inhibition of colony formation in A549 and NCI-H292 cells treated by KWM. (**D**) Cells were treated with the indicated doses of KWM, scratch wound assay was performed after treatment. Images were captured by optical microscopy. Data were expressed as mean ± SD, *n* = 3. *** *p* < 0.001 versus control groups.

**Figure 2 ijms-24-01015-f002:**
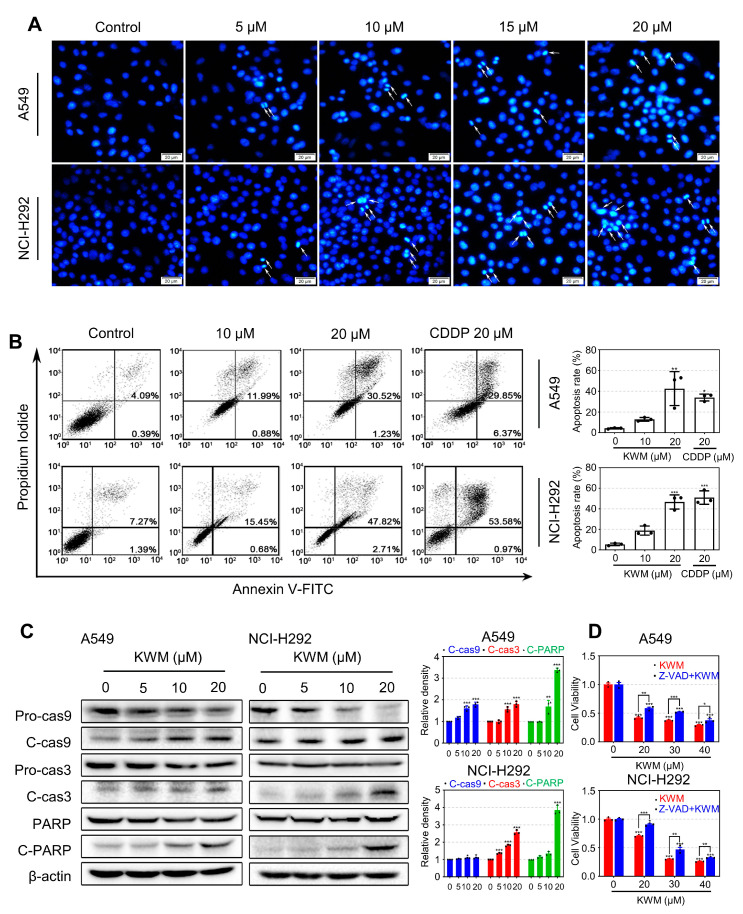
KWM-induced apoptosis in A549 and NCI-H292 cells. (**A**) Cells were incubated with different concentrations of KWM for 24 h, and the changes in nuclear morphology were recorded by fluorescence microscopy after DAPI staining. (**B**) Cells were treated with KWM for 48 h, stained with Annexin V/PI and a flow cytometry assay was performed. CDDP treatment was used as positive control. (**C**) Cells were treated with KWM as indicated doses for 24 h, and cell lysates were subjected to immunoblotting for caspase-3, caspase-9 and PARP. The β-actin was used as the loading control. (**D**) A549 and NCI-H292 cells were pretreated for 2 h with Z-VAD-FMK (20 μM) prior to addition of KWM or DMSO. Cell viability was detected by MTT assay after 48 h. The results were mean ± SD from three independent experiments. * *p* < 0.05, ** *p* < 0.01, and *** *p* < 0.001 versus control groups.

**Figure 3 ijms-24-01015-f003:**
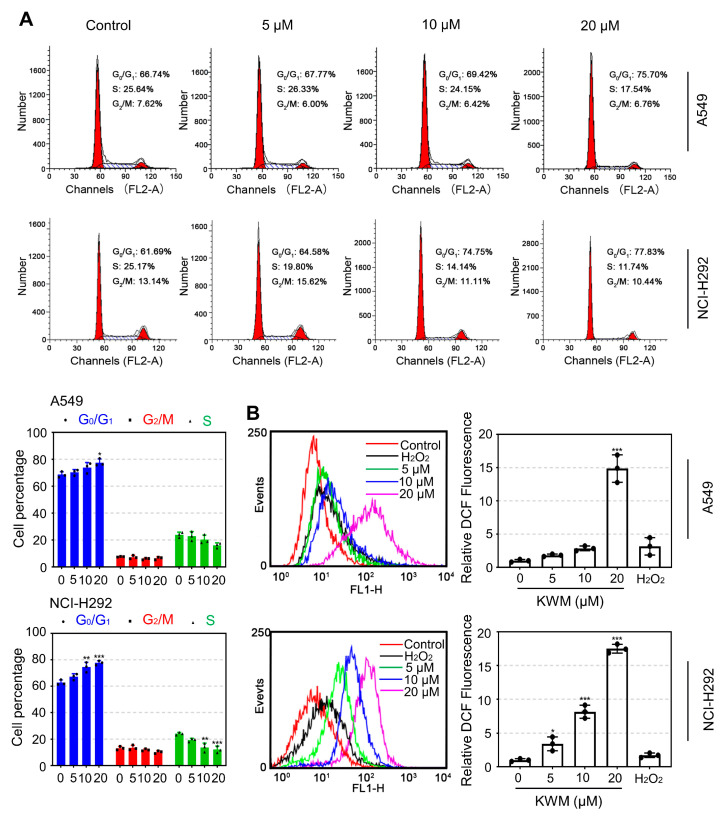
KWM-induced cell cycle arrest and accumulation of ROS in A549 and NCI-H292 cells. (**A**) A549 and NCI-H292 cells were treated with the indicated doses of KWM for 24 h, and PI staining and flow cytometry analysis were used to illustrate the distribution of each phase. (**B**) Cells were treated with KWM for 24 h. The fluorescence obtained by the oxidation of DCFH–DA was determined by flow cytometry. H_2_O_2_ treatment was used as a positive control. The relative fluorescence intensity was presented as mean ± SD of three independent experiments. * *p* < 0.05, ** *p* < 0.01, and *** *p* < 0.001 versus control groups.

**Figure 4 ijms-24-01015-f004:**
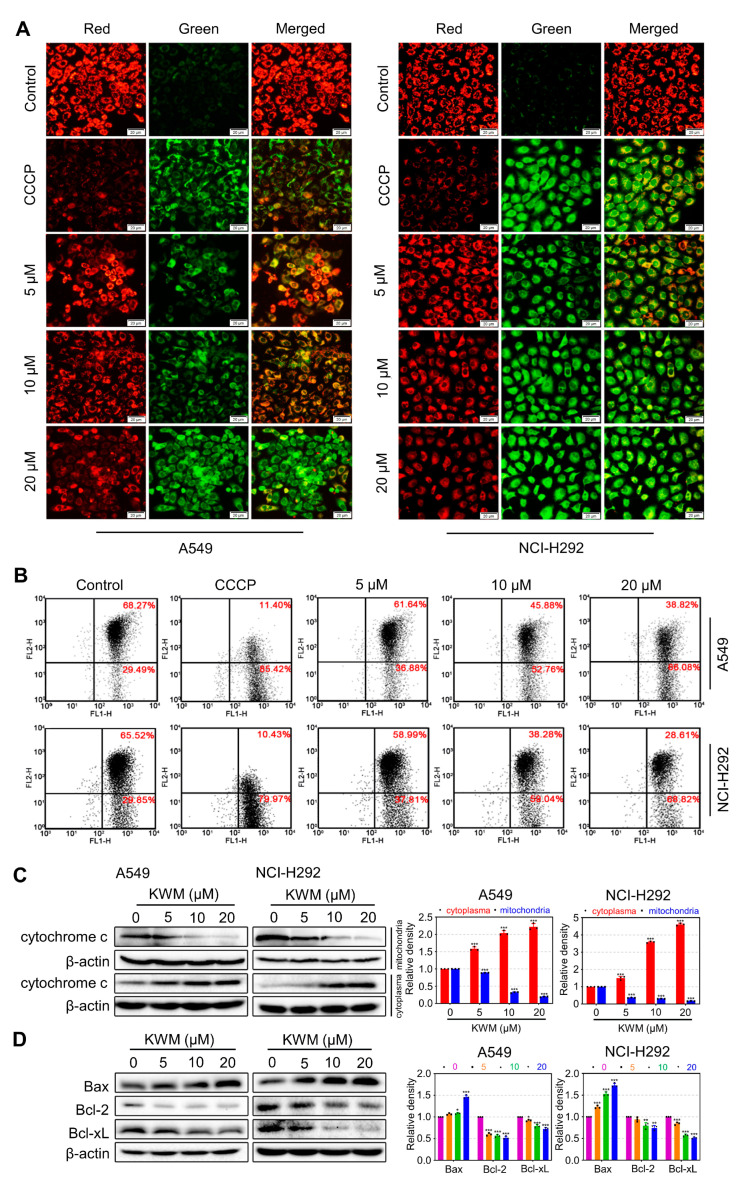
KWM induced apoptosis through the mitochondria-mediated intrinsic pathway in A549 and NCI-H292 cells. (**A**) Cells were incubated with KWM for 24 h. MMP changes were monitored by loading with JC-1 and detected by fluorescence microscopy. (**B**) MMP changes were analyzed by flow cytometry. (**C**) The cytosolic extracts and mitochondrial fraction were analyzed by immunoblotting to detected mitochondrial cytochrome *c* release. (**D**) Cells were incubated with KWM for 24 h, cell lysates were subjected to immunoblotting for Bax, Bcl-2 and Bcl-xL. β-actin was used as a loading control. Values were expressed as mean ± SD, *n* = 3. * *p* < 0.05, ** *p* < 0.01, and *** *p* < 0.001 versus control groups.

**Figure 5 ijms-24-01015-f005:**
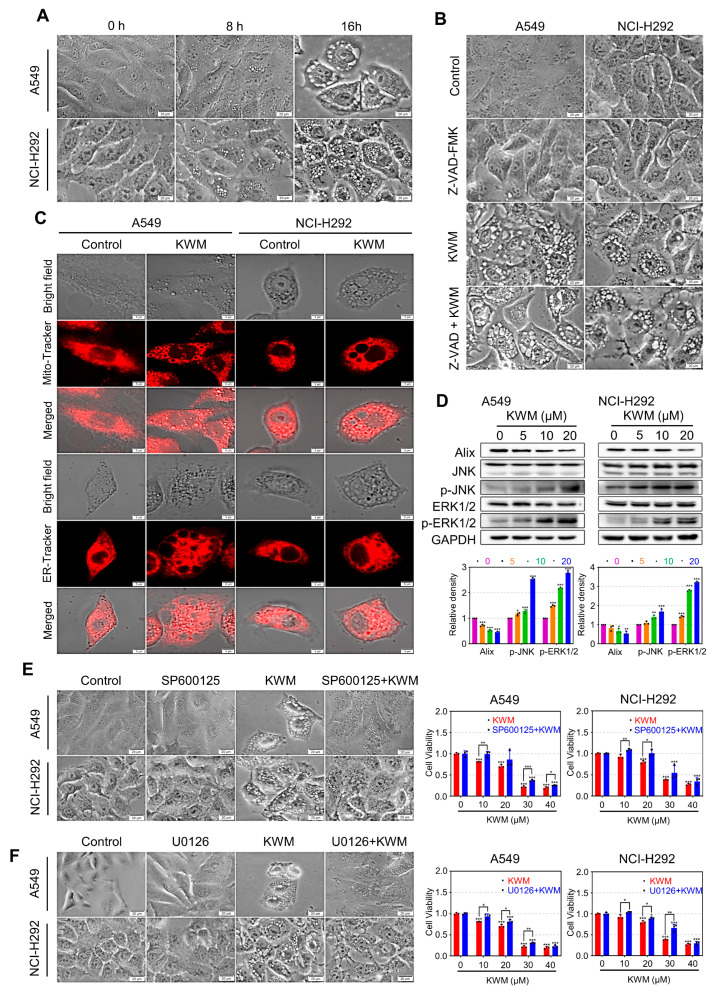
KWM-induced paraptosis in A549 and NCI-H292 cells. (**A**) A549 and NCI-H292 cells were treated with KWM (20 μM), phase-contrast images were taken at indicated time points. (**B**) A549 and NCI-H292 cells were pretreated for 2 h in the presence or absence of Z-VAD(OMe)-FMK (20 μM) prior to addition of KWM (20 μM) or DMSO. Phase-contrast images were captured after 24 h. (**C**) Cells were treated with 20 μM KWM for 24 h, then incubated with 1 μM ER-Tracker Red or 125 nm Mito-Tracker Red for 30 min at 37 °C. Phase-contrast and fluorescent images were captured by confocal microscope. (**D**) Cells were incubated with KWM for 16 h, cell lysates were subjected to immunoblotting for paraptosis associated proteins. (**E**,**F**) Cells were pretreated for 2 h in the presence or absence of SP600125 (10 μM) or U0126 (10 μM) prior to addition of KWM (20 μM) or DMSO. Phase-contrast images were captured after 12 h of treatment. Cell viability was determined by MTT assay after 48 h. Results were from three separate experiments. Values were expressed as mean ± SD, *n* = 3. * *p* < 0.05, ** *p* < 0.01, and *** *p* < 0.001 versus respective control groups.

**Figure 6 ijms-24-01015-f006:**
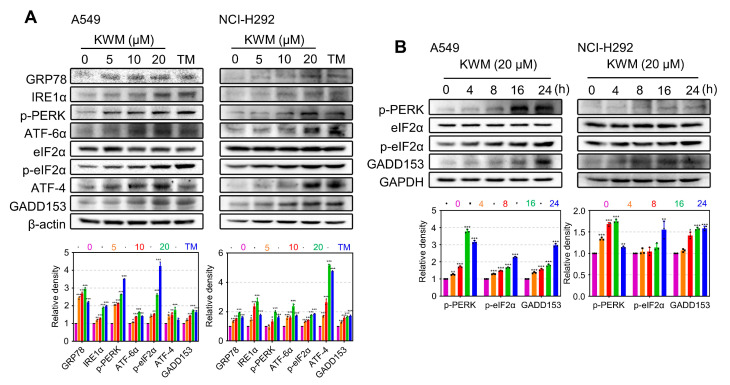
KWM induced ER stress and UPR in A549 and NCI-H292 cells. (**A**) Cells were treated with KWM for 16 h, cell lysates were subjected to immunoblotting for ER stress and UPR-associated proteins. Tunicamycin (TM, 3 μM) treatment was used as positive control. (**B**) Cells were treated with 20 μM KWM for different times, several representative proteins of UPR pathway were detected by immunoblotting. Values were expressed as mean ± SD, *n* = 3. * *p* < 0.05, ** *p* < 0.01, and *** *p* < 0.001 versus untreated groups.

**Figure 7 ijms-24-01015-f007:**
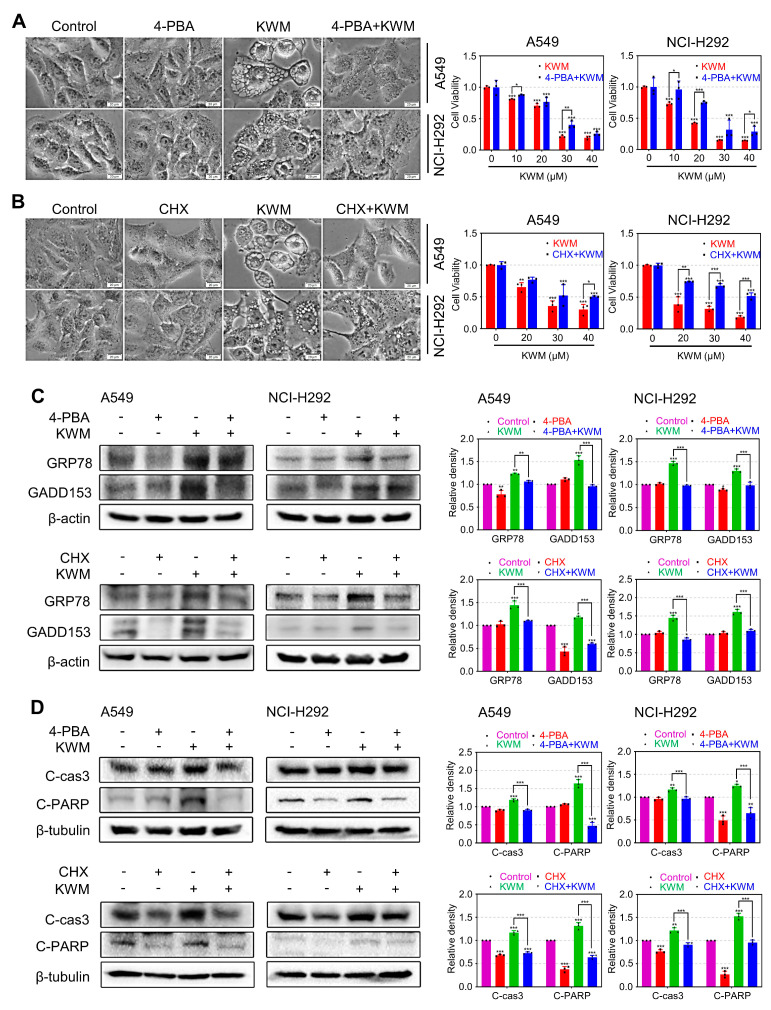
KWM-induced apoptosis and paraptosis mediated by ER stress. (**A**,**B**) Cells were pretreated for 2 h in the presence or absence of 2 mM 4-PBA or 5 μM CHX prior to treatment of KWM (20 μM) or DMSO. Phase-contrast images were captured after 12 h. Cell viability was determined by MTT assay after 48 h. (**C**,**D**) Cells were pretreated with 4-PBA (2 mM) or CHX (5 μM) for 2 h and then treated with or without KWM (20 μM) for additional 16 h (**C**) or 24 h (**D**), cell lysates were subjected to immunoblotting for GRP78 and GADD153, β-actin was used as a loading control (**C**). And cell lysates were subjected to immunoblotting for apoptosis associated proteins, β-tubulin was used as a loading control (**D**). Results were from three separate experiments. Values were expressed as mean ± SD, *n* = 3. * *p* < 0.05, ** *p* < 0.01, and *** *p* < 0.001 versus respective control groups.

**Figure 8 ijms-24-01015-f008:**
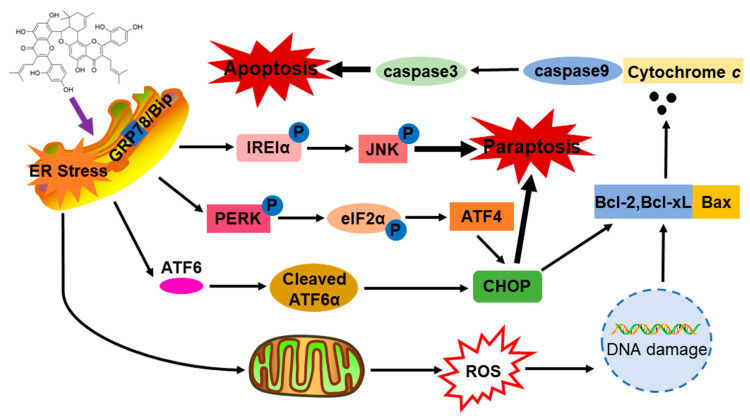
The working model of KWM.

## Data Availability

Not applicable.

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
