# Peer review of "A Mulberry Diels-Alder-Type Adduct, Kuwanon M, Triggers Apoptosis and Paraptosis of Lung Cancer Cells through Inducing Endoplasmic Reticulum Stress"

_ijms, 2023, doi:10.3390/ijms24021015_

Round 1

Reviewer 1 Report

The NMR data should be added to the Materials section. The rules behind the given order of the NMR signals is unclear.

It is unclear how the authors have isolated KWM which was used in this study. Please provide more information.

The authors should describe how they have established the shown structure of KWM according to the mass and NMR spectra.

The described IC50 values around 10 µM are rather moderate. I suppose cisplatin is much more active in these cancer cells. Please explain why KWM deserves publication in this journal since the compound is not new.

Figure 1: Data of a positive control (approved anticancer drug, best a drug applied against lung cancer or cisplatin which was used as control in other experiments of this study) must be provided in order to see how well KWM performs when compared with an established anticancer drug.

Reviewer 2 Report

In this manuscript, the authors studied the growth inhibitory effects of Kuwanon M (KWM), which was isolated from the root bark of Mulberry trees(Morus alba),  on human lung cancer cells and related possible mechanisms. 

I am overall satisfied with the research design of the manuscript but my serious concern is for the statistical analysis. 

1. In most of the figures, the significance was shown without specifically indicating which comparison it is. Fig 1B, 1D, 2B, 2D, and so on. Basically, every figure with statistical analysis has this problem. The authors need to make it clear which comparison is significant.

2. In all the statistical analyses, there were 3 replicates, and the authors used the ANOVA test and post hoc multiple comparison Bonferroni test and got p-values less than 0.001 for many of the comparisons. It is hard to believe that one can get such a high level of significance power using such few numbers of samples. I am suggesting the author double-check the statistical analyses and provide your original raw data as supplemental material. It would also be great if you can plot the 3 replicates dots with the barplots.

Minor points:

1. There are many English language and style issues needed to fix. Only on the first page, L9 and L27, "Mulberry tree (Morus alba) have been", "have" should be "has"; L28, "has been" should be "have been"; L39, "one of the 38 most common and deadliest type of cancers", "type of cancers" should be "types of cancer"; 

2. The figures are all sort of blurred. 

Reviewer 3 Report

In the present manuscript, Mengjiao Ma and co-authors analyzed the inhibitory effects of KWM on human lung cancer cells and the induction of apoptosis and paraptosis, as a potential lung cancer therapeutic agent. The study is very interesting, the paper well written and results clearly presented. However, there are some revisions needed before accepting for publication. 

It is important to highlight that this kind of molecules have an antiproliferative effect only against cancer cells and no other healthy cells (see, e.g., doi: 10.1371/journal.pone.0192178; doi: 10.3390/molecules23112922). Did the authors test the same compounds under the same conditions on healthy cells?

Figure 1 panel C is very difficult to see. Please provide bigger and high-resolution images.

Figure 1 panel D, please provide high resolution images.

Figure 2 panel A, figures lack of scale bar. 

Figure 3 panel A, panels are difficult to read. Please provide bigger and high-resolution images.

Figure 4 panel A, figures lack scale bar. Please modify and provide high resolution images.

Figure 5 panel C figures lack scale bar.

Round 2

Reviewer 1 Report

The revised manuscript is suitable for publication now.

Author Response

Thank you!